# Challenges and Perspectives of Mapping Locus Coeruleus Activity in the Rodent with High-Resolution fMRI

**DOI:** 10.3390/brainsci12081085

**Published:** 2022-08-16

**Authors:** Xiaoqing Alice Zhou, Yuanyuan Jiang, Vitaly Napadow, Xin Yu

**Affiliations:** 1Athinoula A. Martinos Center for Biomedical Imaging, Massachusetts General Hospital, Charlestown, MA 02129, USA; 2Department of Physical Medicine and Rehabilitation, Spaulding Rehabilitation Hospital, Harvard Medical School, Charlestown, MA 02129, USA

**Keywords:** locus coeruleus, fMRI, rodent brain, brainstem, single-vessel fMRI

## Abstract

The locus coeruleus (LC) is one of the most commonly studied brainstem nuclei when investigating brain–behavior associations. The LC serves as a major brainstem relay for both ascending bottom-up and descending top-down projections. Specifically, noradrenergic (NA) LC neurons not only connect globally to higher-order subcortical nuclei and cortex to mediate arousal and attention but also directly project to other brainstem nuclei and to the spinal cord to control autonomic function. Despite the extensive investigation of LC function using electrophysiological recordings and cellular/molecular imaging for both cognitive research and the contribution of LC to different pathological states, the role of neuroimaging to investigate LC function has been restricted. For instance, it remains challenging to identify LC-specific activation with functional MRI (fMRI) in animal models, due to the small size of this nucleus. Here, we discuss the complexity of fMRI applications toward LC activity mapping in mouse brains by highlighting the technological challenges. Further, we introduce a single-vessel fMRI mapping approach to elucidate the vascular specificity of high-resolution fMRI signals coupled to LC activation in the mouse brainstem.

The locus coeruleus (LC) is one of the most commonly investigated brainstem nuclei to explore brain–behavior associations [1,2,3]. Studies of LC noradrenergic (NA) pathway-mediated arousal can be traced back half a century [4], originating with investigations of the LC contribution to rapid eye movement (REM) sleep [5,6]. Although LC is a relatively small nucleus, it includes condensed NA neurons with broad projections throughout the central nervous system [7,8]. Such cellular specificity allows for effective stimulation with optogenetic or chemogenetic tools by introducing optical fibers or local microinjection of DREADD agonists [9,10] in rodent models. However, since LC is located deep within the dorsal pontine area of the brainstem, this nucleus remains difficult to image with conventional optical imaging methods. In vivo electrophysiological recording is typically used to measure LC-specific NA neuronal activity based on unique firing patterns, which can guide the precise localization of electrode placement [11]. However, NA influence is widespread in the brain, and the cortical and subcortical neuromodulatory effects of LC firing have been difficult to assess. To bridge LC-specific NA activation and its brain-wide neuromodulatory effects, functional MRI (fMRI) has been applied to explore LC activity and influence on other brain regions for both animal and human brains [12,13]. In this review, we mainly focus on elucidating the challenges of applying fMRI methods to precisely target LC neuronal activity in the rodent brain.

*Can fMRI accurately assess LC activity in the mouse brain with sufficient resolution?* The mouse LC has a cross-sectional diameter of ~300 µm and a slightly longer dimension along the rostral–causal brainstem axis [14]. Conventional application of fMRI has used a two-dimensional (2D) echo-planar imaging (EPI) pulse sequence approach. Typical 2D EPI-fMRI imaging of the mouse brain can be performed with an in-plane resolution of 200 × 200 µm and 500 µm slice thickness, e.g., awake mouse fMRI using a 14 T scanner [15]. Hence, given this spatial resolution, the mouse LC would be typically comprised of fewer than four voxels and notable partial volume effects of these voxels due to contributions from adjacent nuclei. Moreover, for group analysis, cross-subject coregistration methods contribute additional smoothening effects for the signal from the LC, further reducing LC anatomical specificity of the fMRI signal. It should be noted that the resting-state functional connectivity mapping related to the LC-NE pathway mainly encompasses major whole-brain resting-state networks [9,10] and may not reflect LC-specific low-frequency signal fluctuations, due to a lack of sufficient spatial resolution with the conventional fMRI methods noted above.

To improve LC spatial specificity in fMRI, a better appreciation of the causes of limited spatial resolution of fMRI is needed. The first issue includes limits of the signal-to-noise ratio (SNR) for task-related and resting-state fMRI using conventional mapping methods, e.g., echo-planar imaging (EPI), to acquire time-series datasets with a 1 or 2 s sample rate, i.e., the time of repetition (TR). Smaller voxel size is associated with a lower SNR due to a reduced number of water protons in the voxel. The second issue is the strength of the MRI gradient and its slew rate (i.e., how fast to approach the peak amplitude of the gradient) needed to yield high spatial resolution fMRI across the mouse brain at a selected time of echo (TE). TE is the free induction decay (FID) time of water spins following radiofrequency (RF) pulses to acquire a T2*-weighted signal, which represents the change in field homogeneity due to neuronal activity-coupled hemodynamic responses, e.g., the blood oxygen level-dependent (BOLD) fMRI signal [16,17,18,19]. The neuronal activity-coupled T2* contrast changes vary at different TE, which can be influenced by the intrinsic susceptibility of the samples (blood vs. parenchyma voxels), the field strength-dependent T2* of brain tissue, and the echo-train duration during acquisition [20,21,22,23]. To optimize the T2* contrast for a small nucleus like the LC, we need to optimize the TE and echo acquisition, and improved gradient performance is critical for such high spatial resolution data acquisition.

To solve these technological challenges, ultrahigh-field preclinical MR scanners (>11.7 Tesla) and multiarray cryoprobe [24] or implanted RF coils [25] have been applied to boost the SNR, while high-performance gradients (e.g., 1 T/m gradient strength) can be coupled with an ultrahigh-field scanner to enable high spatial resolution mapping [26,27]. Moreover, several advanced denoising algorithms [28] have been developed to remove background noise and further improve the SNR of high-resolution fMRI images [29]. Ultimately, using a 14 Tesla MRI scanner, the spatial resolution of mouse brain fMRI can approach 100 um isotropic [30], a substantial improvement in spatial resolution, making it possible to record fMRI signals from dozens of LC voxels. Additionally, it should be noted that the LC in human brains is much larger than in mice. The human LC is ~14.5 mm long and 2–2.5 mm in cross-sectional diameter for this nucleus located 1–1.5 mm away from the floor of the fourth ventricle [31], a high source of noise due to CSF fluctuations. For human MRI, the ultrahigh field is typically 7 T and above, with fMRI data from 10.5 Tesla scanners approaching 400 um isotropic resolution, allowing for better localization of LC [29,32]. Such promising advances in high-resolution human fMRI could lead to significant advances in human LC research.

*What is the spatial specificity of fMRI to map LC activation?* It has been well understood that fMRI signals, e.g., typical BOLD [16] or cerebral blood volume (CBV; [33] contrasts, originate from blood-based contrast mechanisms and are sensitive to blood vessel density and physiology. In contrast to human fMRI using 3 T or 7 T scanners, rodent fMRI is typically performed using specialized preclinical MRI scanners operating at field strengths above 9.4 T. Using ultrahigh-field strength, e.g., 14 T, to improve the spatial resolution of fMRI to 100 × 100 um in-plane, Yu et al. recently reported that individual penetrating venules dominated BOLD signals because of the high oxygen:hemoglobin (Hb) ratio changes in venous blood, while arterioles dominate CBV signals due to the smooth muscle cell-controlled arterial dilation in both deep cortex [26,34] (Figure 1) and hippocampus [35]. These microscale vessels have 20–70 micron diameter and are separated by 300–400 microns, connecting large draining veins or feeding arteries to higher-order vascular branches toward capillary beds [36,37].

The LC is located in the superior dorsal pontine area, near the fourth ventricle of mouse brains. High-resolution micro-CT has been used to identify the microvasculature of the mouse brainstem, showing the densely packed choroid plexus in the fourth ventricle and penetrating vessels supplying blood to the dorsal–rostral pontine area close to the LC [38] (Figure 2). While there has not been a report of brainstem functional mapping with single-vessel fMRI, it is highly plausible to detect activation-coupled peak hemodynamic responses located at specific penetrating vessels supplying blood to the LC, which could also spread to the choroid plexus. The fMRI signal from LC parenchymal voxels would show much less contrast-to-noise ratio (CNR) difference compared to surrounding vessel voxels. This hypothesized technique would extend LC-specific fMRI from a nuclei-specific approach to a vessel-specific spatial pattern approach, consistent with the vascular origin of fMRI.

To better differentiate the vascular contribution to LC spatial specificity with high-resolution fMRI, it is important to specify intravascular versus extravascular effects on T2*-weighted signals to a given voxel [39]. The vascular volume fraction of brain tissue is usually less than 2–5% [40,41,42]. When the voxel size (e.g., 1 mm isotropic) is significantly larger than the vessel size, the intravascular signal contribution to the overall T2* signal in the voxel is small, given its volume fraction. This vascular effect would be more negligible at high magnetic field strength because of the faster T2* decay of venous blood, despite its higher oxy/deoxygenation ratio changes compared to other compartments in the voxel. Also, the extravascular influence of blood can be further reduced by a spin-echo sequence, which introduces a 180-degree radiofrequency pulse to resume additional dephasing during the free induction decay caused by the static field distortion of larger vessels [43,44,45]. It should be noted that fMRI signals of large voxels weigh more on capillary beds, which are closer to the source of parenchymal neuronal activation. In contrast, when the voxel size is smaller (e.g., 100 um isotropic), the penetrating vessel, which is tens of microns in diameter, presents a much higher volume fraction, and its vascular contribution to the T2*-weighed fMRI signal would not be negligible, but may even dominate the BOLD or CBV effects, as previously reported by Yu et al. [20], Thus, when the spatial resolution of fMRI is further improved, a more refined vessel-specific fMRI signal may be evident across brainstem functional nuclei, including LC.

*What does the future of high-resolution LC fMRI promise?* While the role of neuroimaging to investigate LC function has been restricted, mainly due to technical challenges in identifying LC-specific activation with fMRI, these challenges may be overcome to yield a more robust recording of fMRI signals from the LC. The application of LC-specific high-resolution fMRI mapping could lead to multiple novel insights relevant to LC-NA mediated neuromodulatory function. First, most fMRI studies use general linear models (e.g., linear regression) for analysis of the fMRI signal, incorporating a canonical hemodynamic response function (HRF). This HRF relies on the underlying neurovascular coupling features of local brain regions, which are based on the cellular components and vascular structure within the parenchyma. Different HRFs across subcortical nuclei have been reported, for instance for hippocampal activity in monkey brains [46]. In fact, the estimation of a regionally specific HRF relies not only on voxel location but also on voxel size. High-resolution fMRI could provide a more precise estimation of the vessel-specific HRF located at the LC and distinguish it from spreading responses associated with the choroid plexus of the fourth ventricle. This LC-specific vascular hemodynamic mapping could enable the comparison of neurovascular coupling effects and HRFs between the LC and other cortical and subcortical structures. Also, high-resolution fMRI may allow for the ability to distinguish LC-coupled vascular responses from activation of adjacent functional nuclei, e.g., the parabrachial (PB) nucleus [47,48]. Although similar brainstem-penetrating vessels may supply blood to both nuclei, the vessel-specific dynamic patterns could be characterized if LC- or PB-specific neural circuit activity is manipulated with optogenetic or chemogenetic tools. Moreover, high-resolution fMRI will provide a unique neuroimaging platform to map cross-scale vasodynamics mediated through the LC-NA pathway. Either NA-mediated vasoconstriction or intrinsic dynamic vasomotor changes could be directly mapped with a single-vessel fMRI technique, in combination with LC-specific vascular dynamic changes.

As one of the most commonly studied brainstem nuclei, the LC is critical for a wide array of brain–behavior associations. Given its tiny size and unique spatial localization in the brainstem, how to precisely locate LC function in the context of cross-scale brain dynamic mapping is an ongoing challenge. High-resolution awake mouse fMRI, in particular with ultrahigh field strength, in combination with other optogenetic or chemogenetic techniques will yield insights into deciphering the LC-NE circuitry contribution to more complex arousal and attention regulation. Meanwhile, other emerging imaging techniques, e.g., functional ultrasound (fUS)and optoacoustic (fOA) imaging, have been developed for large-scale high spatiotemporal resolution rodent brain imaging [49,50,51,52,53,54]. In particular, fUS is also adept at detecting vessel-specific vasodynamic changes, which mimic single-vessel fMRI results when sufficient SNR is present in deep brain regions, e.g., the LC of awake mice. It should also be noted that to achieve optimal signal transmission for fOA or fUS, rodents need to have either a thin skull or a large craniotomy needs to be performed. However, both fOA and fUS are characterized by more efficient spectral transmission through brain tissue than conventional optical imaging methods. In summary, the LC is a critically important brainstem nucleus, and newer imaging modalities are beginning to show promise for capturing functional activity from the LC for many research applications.

## Figures and Tables

**Figure 1 brainsci-12-01085-f001:**
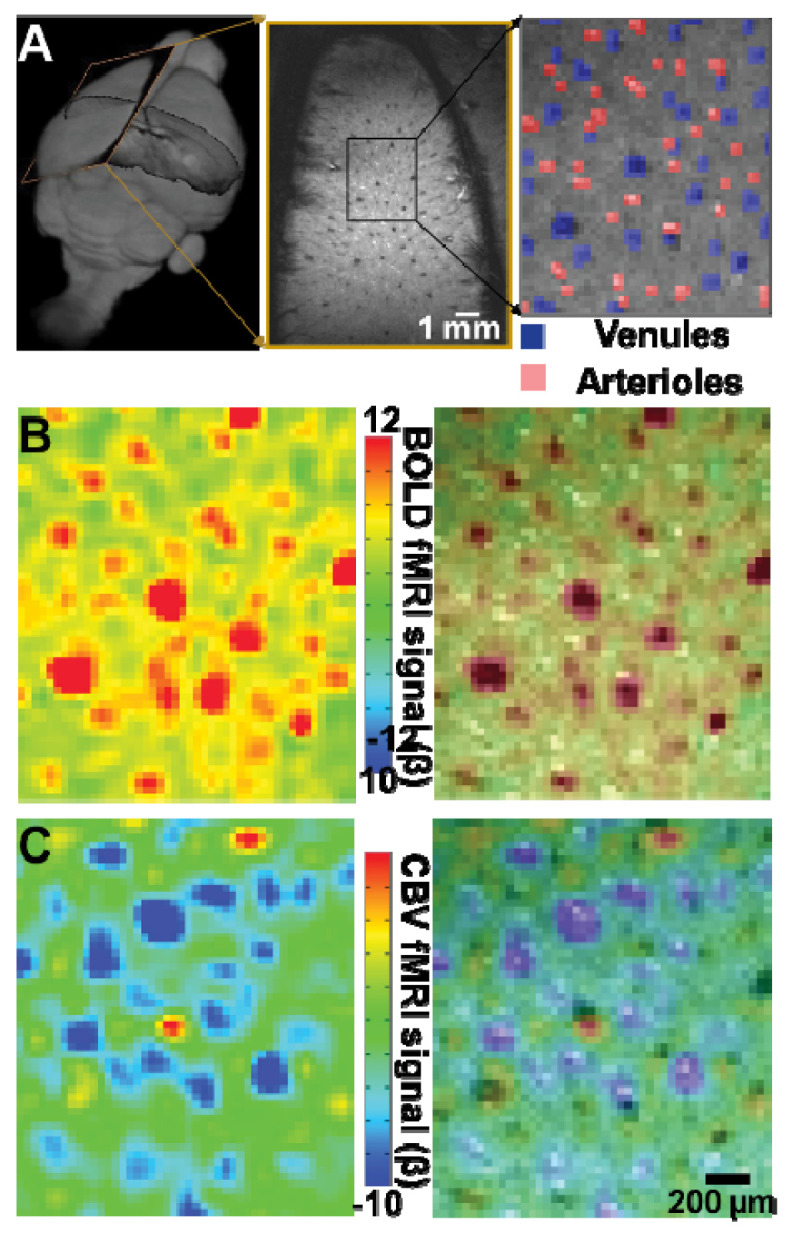
Vessel-specific BOLD and CBV signals with single-vessel fMRI. (**A**). 2D multigradient-echo (MGE) slice is acquired across the deep-layer cortex to identify penetrating arterioles (red) and venules (blue). (**B**). The evoked BOLD fMRI signal of the forepaw somatosensory cortex (FP-S1) is located at the penetrating venules. (**C**). The evoked CBV fMRI signal of FP-S1 is located at the penetrating arterioles. Adapted from Yu et al. 2016.

**Figure 2 brainsci-12-01085-f002:**
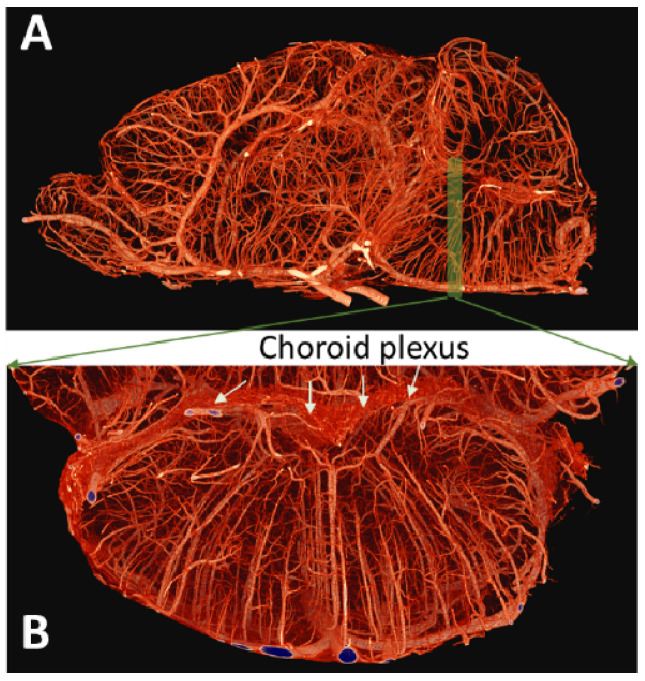
High-resolution micro-CT-based cerebrovascular mapping in the mouse brainstem. (**A**) Sagittal view and (**B**) coronal view of vascular structure supplying the superior dorsal pontine area), adapted from Hlushchuk et al. 2020.

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
