# Peer review of "Challenges and Perspectives of Mapping Locus Coeruleus Activity in the Rodent with High-Resolution fMRI"

_brainsci, 2022, doi:10.3390/brainsci12081085_

Round 1
Reviewer 1 Report
This is interesting small opinion paper on timely topic. Overall manuscript is well written and provides insight to technical challenges related to detection of fMRI changes in locus coeruleus. Discussion is also relevant for fMRI of other similar small nuclei and their relation to network activity. I have only some minor comments that authors may want to consider to further improve the clarity of the writing in a few points.
1) 3rd paragraph of Intro. Sentence "Smaller voxel size is associated with lower SNR due to a reduced number of water protons in the voxel at a given time, i.e., the time-of-repetition (TR), which is usually set at 1 or 2 s for
fMRI" is somewhat confusing. It gives an impression that amount of water protons within voxel is dependent on repetition time. I don't think this is true, but authors probably are just referring to smaller voxel containing less water spins.
2) 3rd paragraph of Intro. Sentence "Thus, an optimal gradient performance is needed to optimize TE for high spatial resolution data acquisition."
Maybe better say for example "...to achieve optimal TE for high spatial resolution" Also, there could be a bit more elaboration what is the optimal TE - also mentioning that it gets short in high Bo field which is needed to high SNR (especially as many examples later on are in ultrahigh filed MRI in 14 T)
3) Please mention field strength when discussing results from Yu et al 2016 in the beginning of paragraph "What is the spatial specificity of fMRI to map LC activation?" This is very relevant information when discussing venous vs arterial contribution to fMRI contrast.
4) The third last paragrah starting "The paradox to improve LC spatial specificity with high-resolution fMRI results from the intravascular versus extravascular..." may benefit from some rewriting. Extravascular contribution is not really never explained, and in current form storyline might be a bit difficult to follow for a non-expert reader.
Author Response
We thank the reviewer for valuable comments. We have made point-to-point responses, which have been summarized in the following:
- 3rd paragraph of Intro. Sentence "Smaller voxel size is associated with lower SNR due to a reduced number of water protons in the voxel at a given time, i.e., the time-of-repetition (TR), which is usually set at 1 or 2 s for fMRI" is somewhat confusing. It gives an impression that amount of water protons within voxel is dependent on repetition time. I don't think this is true, but authors probably are just referring to smaller voxel containing less water spins.
We have revised this sentence to improve the clarity.
2) 3rd paragraph of Intro. Sentence "Thus, an optimal gradient performance is needed to optimize TE for high spatial resolution data acquisition."Maybe better say for example "...to achieve optimal TE for high spatial resolution" Also, there could be a bit more elaboration what is the optimal TE - also mentioning that it gets short in high Bo field which is needed to high SNR (especially as many examples later on are in ultrahigh filed MRI in 14 T)
As suggested by the reviewer, we have discussed the TE-dependent effect on fMRI contrast, including the high field-related TE effect.
3) Please mention field strength when discussing results from Yu et al 2016 in the beginning of paragraph "What is the spatial specificity of fMRI to map LC activation?" This is very relevant information when discussing venous vs arterial contribution to fMRI contrast.
We have added the field strength requirements for high spatial resolution fMRI at the beginning of this paragraph.
4) The third last paragrah starting "The paradox to improve LC spatial specificity with high-resolution fMRI results from the intravascular versus extravascular..." may benefit from some rewriting. Extravascular contribution is not really never explained, and in current form storyline might be a bit difficult to follow for a non-expert reader.
We have revised this sentence.
Reviewer 2 Report
This paper discussed the challenges and opportunities of mapping locus coeruleus in rodents with high-resolution fMRI. Due to the small size of LC, whether fMRI is a promising tool for studying LC in rodents is still an open question.
11) The authors mentioned Hang et al used 2D EPI-fMRI to achieve an in-plane resolution of 200x200 um. However, I couldn’t find Hang’s paper in the References. Has it been included?
22) Palmiter’s 2018 paper discussed the PB nucleus. However, it didn’t mention the technology of high-resolution fMRI at all. Could the authors explain why this paper is cited?
33)Functional ultrasound imaging (fUSI) has become an emerging neuroimaging technique because of low-cost and higher spatial resolution compared with fMRI. Therefore, could fUSI be potentially used for imaging rodent brains? How it fUSI compared with fMRI in studying small nuclei in rodents?
Author Response
We thank the reviewer for the valuable comments. Here, we provide the point-to-point responses:
- The authors mentioned Hang et al used 2D EPI-fMRI to achieve an in-plane resolution of 200x200 um. However, I couldn’t find Hang’s paper in the References. Has it been included?
We have added the paper by Hang et al.
- Palmiter’s 2018 paper discussed the PB nucleus. However, it didn’t mention the technology of high-resolution fMRI at all. Could the authors explain why this paper is cited?
We thank the reviewer's suggestion. We have added several reference related to fMRI of PB in animal brains.
- Functional ultrasound imaging (fUSI) has become an emerging neuroimaging technique because of low-cost and higher spatial resolution compared with fMRI. Therefore, could fUSI be potentially used for imaging rodent brains? How it fUSI compared with fMRI in studying small nuclei in rodents?
We thank the reviewer to refer the functional ultrasound (fUS) imaging method. At the end of this manuscript, we have included the alternative methodology including both the fUS and functional optoacoustic (fOA) method's application for high-resolution brain functional mapping.